# Characterization of linear epitope specificity of antibodies potentially contributing to spontaneous clearance of hepatitis C virus

**Asma Ahsan**[1☯], **Saira Dar**[1☯], **Fareeha Hassan**[1], **Farkhanda Ghafoor**[2], **Muhammad Haroon Yousuf**[3], **Syed Shahzad-ul-Hussan**[1]*

**1** Department of Biology, Syed Babar Ali School of Science and Engineering, Lahore University of Management Sciences, Lahore, Pakistan, **2** Pakistan Kidney and Liver Institute and Research Center, Lahore, Pakistan, **3** Shalamar Hospital, Lahore, Pakistan

☯ These authors contributed equally to this work.
* shahzad.hussan@lums.edu.pk

## Abstract

### Background

Around 30% of the HCV infected patients can spontaneously clear the virus. Cumulative evidence suggests the role of neutralizing antibodies in such spontaneous resolution. Understanding the epitope specificity of such antibodies will inform the rational vaccine design as such information is limited to date. In addition to conformational epitope targeted antibodies, linear epitope specific antibodies have been identified that are broadly cross reactive against diverse HCV strains. In this study, we have characterized the potential role of three conserved linear epitopes in the spontaneous clearance of HCV.

### Methods

We tested the reactivity of sera from chronic patients (CP) and spontaneous resolvers (SR) with linear peptides corresponding to three conserved regions of HCV envelope protein E2 spanning amino acids 412–423, 523–532 and 432–443 using ELISA. Subsequently, we characterized the dependency of HCV neutralization by the reactive serum samples on the antibodies specific for these epitopes using pseudoparticle-based neutralization assay.

In ELISA most of the CP sera showed reactivity to multiple peptides while most of the SR samples were reactive to a single peptide suggesting presence of more specific antibodies in the SR sera. In most of the HCVpp neutralizing sera of particular peptide reactivity the neutralization was significantly affected by the presence of respective peptide. HCV neutralization by CP sera was affected by multiple peptides while 75% of the HCVpp neutralizing SR sera were competed by the 432 epitope.

### Conclusions

These findings suggest that individuals who spontaneously resolve HCV infection at the acute phase, can produce antibodies specific for conserved linear epitopes, and those

**Data Availability Statement:** All relevant data are within the manuscript and its Supporting information files.

**Funding:** Syed Shahzad ul Hussan, NRPU grant (20-5904/NRPU/R&D/17) from Higher Education Commission of Pakistan, https://www.hec.gov.pk/english/pages/home.aspx, Funding agency did not play any role except providing funding for the study.

**Competing interests:** The authors have declared that no competing interests exist.

antibodies can potentially play a role in the spontaneous viral clearance. The epitope present in the 432–443 region of E2 was identified as the primary neutralizing epitope with potential role in spontaneous viral clearance and this epitope potentiates for the design of immunogen for prophylactic vaccine.

## Introduction

Globally, chronic hepatitis C virus (HCV) infections occur in over 71 million people with a mortality rate of around 400,000 [1–3]. Although the paradigm shift from interferon to direct acting antiviral (DAA) regimens has significantly improved the situation in developed countries to combat the infection as most of these DAAs are highly efficacious [4,5]. However, the global impact of these treatments is limited by several factors. Globally, only a small proportion of the HCV infected individuals has been estimated to have access to DAAs [6–8]. Due to the asymptomatic nature of the infection, many infected individuals in developing countries remain unaware of their infection status to pursue treatment. Moreover, cost ineffectiveness of DAAs and high mutation index of the virus leading to the emergence of escape mutants and drug resistant viral variants are other contributing factors to the limited global impact of these DAAs [9–12]. Therefore, a prophylactic vaccine is required to effectively control the HCV infections. Understanding of the underlying mechanism of immune response particularly details of epitope recognition by neutralizing antibodies (NAbs) would provide guidance in vaccine design perspective.

Around 30% of the HCV infected individuals can spontaneously clear the virus at the acute phase [13,14]. While effective cellular immune response has been described for the successful viral clearance [15], emerging evidence suggest the role of NAbs in such clearance [16–19]. Although the role of NAbs in spontaneous resolution of the HCV infection is well established, very limited information is available about the epitope specificity of such antibodies.

The E2 envelope glycoprotein primarily mediates viral interactions with different cellular receptors at the early stage of viral entry and is the main target for NAbs. The E2 protein incorporates conformational as well as linear epitopes of NAbs. Three of the highly conserved epitopes of cross-reactive NAbs have been mapped in the E2 regions spanning residues 412–423, 432–443 and 523–532, which here we will refer to as the 412-epitope, the 432-epitope and the 523-epitope, respectively (S1 Fig). These regions are the part of the CD81 receptor-binding site and are located in the neutralizing face of the E2 structure [20–23]. Most of the known antibodies specific for the 523-epitope recognize this site as a part of conformational epitope. However, NAbs specific for the 412- and the 432-epitopes either recognize these sites as linear epitopes or as a part of conformational epitopes [24]. Some of the most potent antibodies that can neutralize a broad range of circulating HCV strains are targeted against these regions and are linear epitope specific [25,26]. At least one of these linear epitope specific antibodies has shown protection in Chimpanzees [27]. On the basis of the demonstrated role of linear epitopes to elicit broadly neutralizing antibodies we hypothesized that linear epitope specific antibodies can have potential role in spontaneous clearance of HCV. Here, we characterized the conserved linear epitope specificity of antibodies present in the sera of chronic patients and spontaneous-resolvers to decipher the details of neutralizing epitopes important for spontaneous resolution of the infection.

## Materials and methods

### Blood collection and serum separation

Ethical Committees of Lahore University of Management Sciences, Pakistan Kidney and Liver Institute and Shalamar Hospital approved the study with reference number: LUMS-IRB/2017-01-12. After the approval and written informed consents, we collected blood samples from 84 individuals including 30 CP, 49 SR and 5 healthy donors. Blood was collected in plain vacutainers and subsequently the serum was separated and stored at -20˚C. Spontaneous resolvers were identified on the basis of positive anti-HCV antibody response and negative viral RNA detection twice with an interval of four weeks. These were identified through screening of patients visiting primary care and gastroenterology clinics of Shalamar Hospital and Pakistan Kidney and Liver Institute (PKLI) by performing relevant diagnostic tests. None of these individuals was aware of his/her past HCV infection that had been cleared without pursuing treatment. Chronically infected patients were identified on the basis persistent infection (viral load) for over six months or those who had developed symptoms of early hepatic cirrhosis. Healthy uninfected donors were defined as individuals with no present or past infection of HCV.

### Enzyme Linked Immunosorbent Assay (ELISA)

Solutions of linear peptides were prepared in 50 mM sodium carbonate–bicarbonate buffer (pH 9.6) at final concentrations of 5 µg/ml. ELISA plates (Thermo scientific Immulon 4HBX) were coated with 100 µl of peptide antigens and incubated overnight at 4˚C. Next day plate surface was blocked using 200 µl of blocking buffer (5% skimmed milk/BSA) after removing peptide solution, incubated at room temperature for 2 hours and washed with 200 µl of washing buffer (0.02% sodium azide in PBS). Primary monoclonal antibody, Ap33 as well as patient sera were diluted in PBS, 50 µl of which was added per well in triplicates and the plate was incubated at room temperature for 1 hour. The AP33 antibody was kindly provided by Dr Arvind Patel, Institute of Infection, Immunity and Inflammation, University of Glasgow. BSA-coated wells and blank wells containing only carbonate buffer served as negative controls for nonspecific binding. Plate was washed and incubated for 1 hour with goat anti-human alkaline phosphatase conjugated IgG (Santa Cruz, sc-2454) for human serum samples and goat anti-mouse alkaline phosphatase conjugated IgG (Abcam, ab97020) as secondary antibodies at 1000-fold dilution. Absorbance was measured at 405 nm on Synergy HTX reader after adding p-nitrophenyl phosphate substrate [28,29]. Data was plotted using the Graphpad prism software.

### HCV neutralization assay

HCV neutralization of serum samples was evaluated using HCV pseudoparticles (HCVpp) expressing luciferase reporter gene as previously described [30]. In brief, HEK293T cells seeded for 24 hours in T-75 flask were co-transfected with 6 µg of HCV envelope expressing vector and 18 µg of pNL4-3.Luc.R⁻ E⁻ backbone vector also containing luciferase reporter gene using lipofectamine-3000 as a transfection reagent. The HCV genotype-1a (strain H77) envelope-expressing vector was kindly provided by Dr Parizia Farci at NIAID, NIH, USA while the backbone vector was obtained through the AIDS Research and Reference Program, Division of AIDS, NIAID, NIH. VSV-G-enveloped plasmid was used to produce VSVpp as nonspecific control. Supernatant containing pseudoparticles was harvested on day 5 after transfection, centrifuged at 2000 rpm, filtered through a 0.45 µm filter and stored at -80˚C. For neutralization, Huh7.5 cells ($1.5 \times 10^4$ cells per well) were seeded into a 96-well plate in 100 µl of growth medium and incubated overnight at 37˚C and 5% $CO_2$. Serum samples were heated at 56˚C

for 30 minutes followed by centrifugation at 1200xg. From the supernatants, serum dilutions were prepared in cDMEM media and incubated with HCVpp for 1.5 hours. Subsequently, serum-HCVpp mix was transferred to wells containing Huh7.5 cells and media was changed after 6 hours. After 72 hr of infection media was removed and cells were lysed by adding 50 μL of 1x GloLysis buffer (Promega). Cell lysate was transferred to white opaque plates and luminescence was measured by adding 45 μL of luciferase substrate (Promega). Percent neutralization of each sample was calculated by comparing with the control containing no serum and $ED_{50}$ values were calculated from the dose response curve.

### Peptide competition assay

100 μl of peptide stock solution (125 μg/ml) was incubated with 100 μl of serum dilution at 37˚C for 2 hours on gentle rocking. The mix was centrifuged at 15000 rpm, 37˚C for 10 min, and supernatant was collected. 180 μl of supernatant was mixed with 150 μl media containing HCVpp and incubated for 1 hour. HCV neutralization assay was repeated as discussed in the previous section [31].

### Statistical analysis

One-way ANOVA with Bonferroni corrections was used to compare HCV neutralization by peptide treated serum samples with untreated ones and significance was defined as $^*p \leq 0.05$, $^{**}p \leq 0.01$, $^{***}p \leq 0.001$ $^{****}p \leq 0.0001$.

## Results

### Linear epitopes to delineate the reactivity of serum samples

To delineate the linear epitope specificity of antibodies in the sera of chronic patients (CP) and spontaneous-resolvers (SR), peptides corresponding to three highly conserved regions of E2 spanning amino acids aa 412–423, aa 523–532 and aa 432–443 were synthesized in consensus sequence of all major genotypes of HCV (Table 1) as we were unaware of the viral genotype that had infected spontaneous resolvers. In some of the peptides with poor predicted solubility, more polar residues were added at the termini to increase their solubility index.

### Reactivity of serum samples to conserved linear epitopes

We used ELISA to profile the reactivity of antibodies in the sera of 84 individuals including 30 CPs, 49 SRs and 5 healthy donors with peptides corresponding to three conserved linear epitopes. In this regard, reactivity of every serum was tested at three different dilutions, 10-fold, 100-fold and 1000-fold. Cutoff was defined as mean optical density (OD) at 405 nm plus 3x the standard deviation from five healthy serum samples. At 10-fold dilution the reactivity of CP sera for the 432-, the 412- and the 523-epitopes were observed in 63%, 43% and 30% of the samples, respectively (Fig 1a). However, at 100-fold dilution only 1 out of 30 samples was

**Table 1. Details of peptides with amino acid sequences corresponding to conserved linear epitopes present in the E2 envelope protein.**

| Epitope name | Length (aa) | Amino acid sequence | Sequential position in E2 |
|---|---|---|---|
| 412 | 16 | *KK*QLVNTNGSWHIN*KK** | 412–423 |
| 432 | 14 | SLNTGFIAGLFY*KK** | 432–443 |
| 523 | 14 | **R*SGAPTYSWGAN*KK** | 523–532 |

* The residues at the termini written in italic letters were added to increase the solubility of the peptide but these are not the part of viral protein sequence.

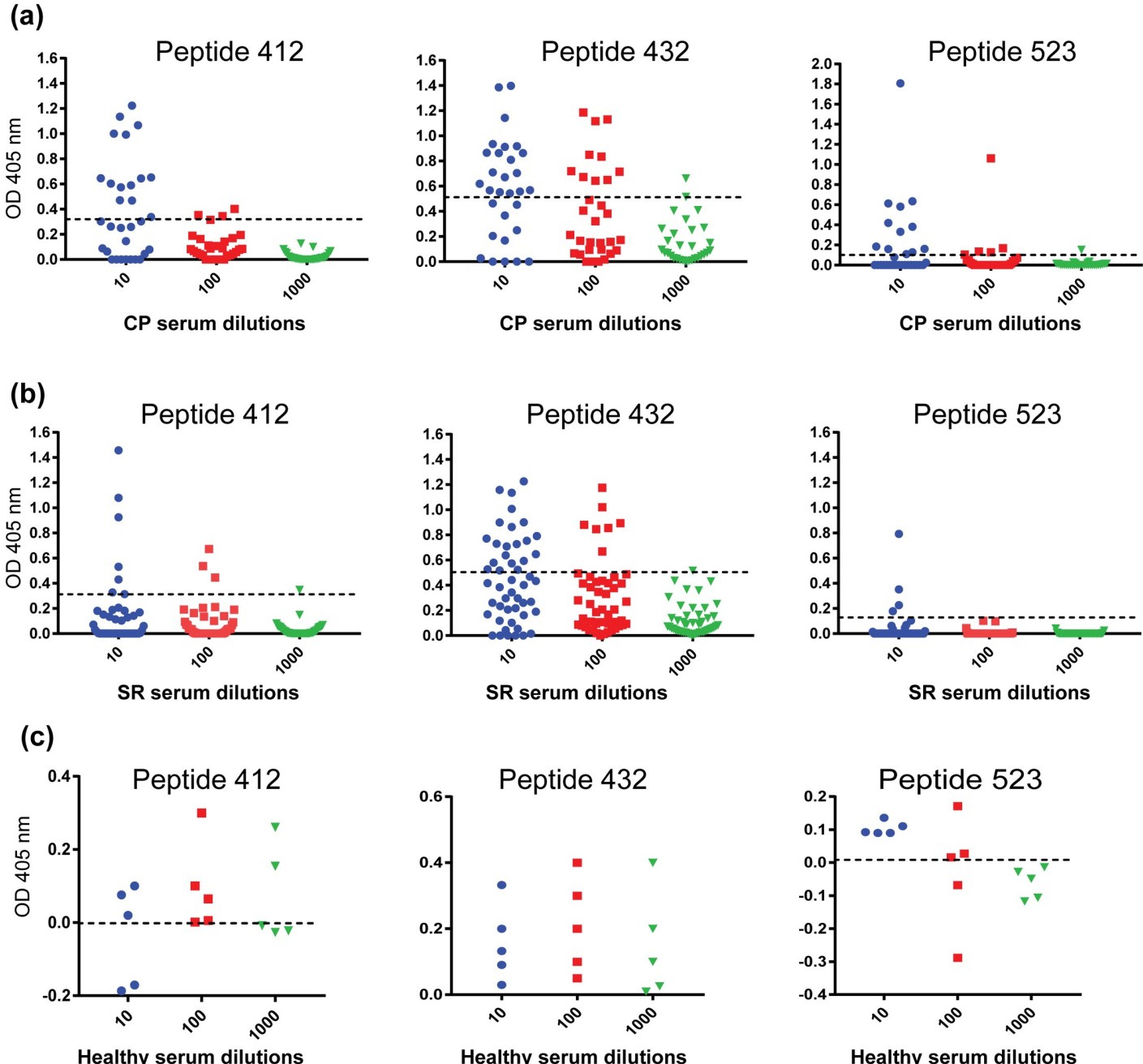

**Fig 1. ELISA-based reactivity of three peptides corresponding to conserved linear epitopes, with CP and SR serum samples.** ELISA was performed using peptides corresponding to three conserved linear epitopes, the 412-epitope, the 432-epitope and the 523-epitope, against three different serum dilutions (10-fold, 100-fold, 1000-fold) of (a) 30 serum samples from chronic patients and (b) 49 serum samples from spontaneous resolvers. Dotted line represents the cutoff value using the mean value from five healthy serum controls plus three times the standard deviation.

reactive to the 412- and the 523-epitopes while 33% of the samples showed reactivity to the 432-epitope (Fig 1a) suggesting the presence of high titers of the 432-epitope specific antibodies. Most of the CP samples were reactive to multiple epitopes (Figs 1a and 2).

In case of the SR serum samples, the observed reactivity was more specific. Only 47% of the SR samples (23 of 49) showed reactivity to any of the peptides. From these samples 78% were

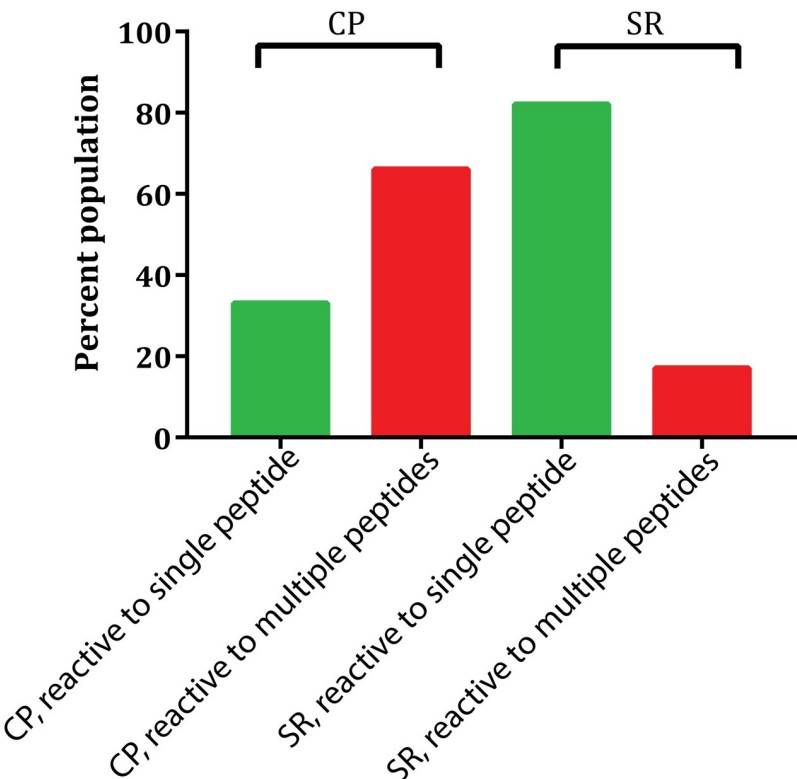

**Fig 2. Comparison of the CP and the SR sera for their reactivity to a single epitope or multiple epitopes.**

reactive to the 432-epitope, while 21% and 17% showed reactivity to the 412- and the 523-epitopes, respectively at 10-fold dilutions. At 100-fold dilutions, reactivity against the 432- and the 412-epitopes was observed in 30% and 13% samples, respectively (Fig 1b). Contrary to the CP samples, most of the positive serum samples from SR group were reactive to a single epitope primarily the 432-epitope– 82% of the reactive SR samples were single epitope specific and 80% of which were specific to the 432-epitope (Figs 1b and 2). This analysis indicated that antibodies reactive to any of these peptides were present only in 47% of the tested SR serum samples while 87% of the CP samples were reactive to these peptides. Secondly, CP sera contained antibodies specific for multiple epitopes while in the SR group most of the samples were reactive to a single epitope.

## HCV neutralization

The serum samples reactive to any of the peptides in ELISA were tested for their ability to neutralize HCV in pseudo-typed virus based neutralization assay. In this regard, initially we used 50-fold serum dilution to obtain preliminary data. Serum samples with more than 30% neutralization in the initial tests were subjected to measure effective dilution with 50% neutralization ($ED_{50}$) through dose response curve. Five serum samples from healthy individuals were also subjected to test their HCV neutralizing activity at 10-fold dilution. None of the healthy serum samples showed any HCVpp neutralization. Moreover, five of the chronic serum samples were tested against VSVpp to rule out the non-specific neutralizing effect. Only negligible neutralizing effect by two of the samples was observed at 10-fold dilution (S2 Fig) suggesting their specificity for HCV. Five of the CP samples showed $ED_{50}$ values higher than 1:500 against

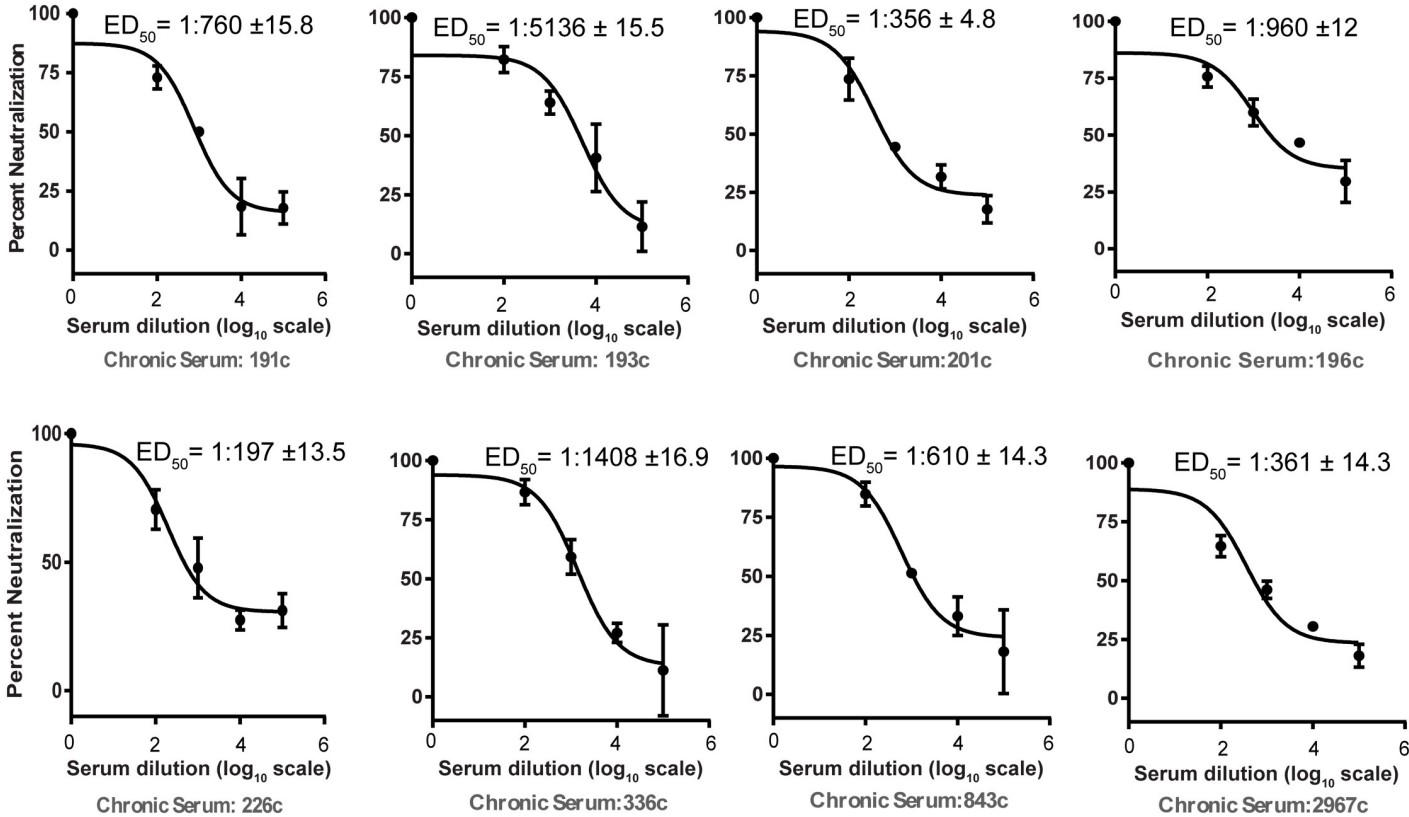

**Fig 3. Dose response curves of selected CP serum samples to measure their $ED_{50}$ values in HCVpp neutralization assay.** Dose response curves showing neutralization of HCVpp by serum samples of chronic patients. Serum samples at varying dilutions were mixed with HCVpp at 37°C and subsequently added to the wells containing Huh7.5 cells. After 72 h, cells were lysed and luciferase activity was measured. Subsequently, percent neutralization of each dilution was calculated by comparing with the control containing no serum.

HCVpp with the most potent sample showing $ED_{50}$ value of 1:5137. Most of these potent CP serum samples had shown reactivity to multiple peptides (Fig 3 and Table 2). Overall, out of 26 reactive CP serum samples 11 neutralized HCVpp with minimum $ED_{50}$ values of 1:75 (Table 2).

**Table 2. Details of peptide reactivity (epitope specificity) and HCVpp neutralization activity of different CP and SR serum samples.**

| Chronic Patients | | | Spontaneous Resolvers | | |
|---|---|---|---|---|---|
| Sample ID | $ED_{50}$ | Epitope specificity* | Sample ID | $ED_{50}$ | Epitope specificity* |
| 191 | 1:760 | 412, 432, 523 | PK2 | 1:222 | 432 |
| 193 | 1:5137 | 412, 432, 523 | PK3 | 1:850 | 432 |
| 201 | 1:356 | 412, 432 | PK13 | 1:16 | 432 |
| 196 | 1:960 | 412, 432, 523 | SH503 | 1:84 | Non reactive |
| 226 | 1:198 | 412, 432, 523 | SH16 | 1:4 | 432, 523 |
| 336 | 1:1408 | 412, 432 | SH511 | 1:212 | 432 |
| 843 | 1:610 | 412 | PK4 | 1:10 | 432 |
| 2967 | 1:361 | 412, 432 | PK7 | 1:113 | 432 |
| 574 | 1:75 | 412, 432 | PK9 | 1:35 | 412 |
| 978 | 1:92 | 432, 412 | PK10 | 1:11 | 432 |
| 762 | 1:127 | 412,432 | PK14 | 1:31 | 412 |
| | | | SH19 | 1:48 | 412 |

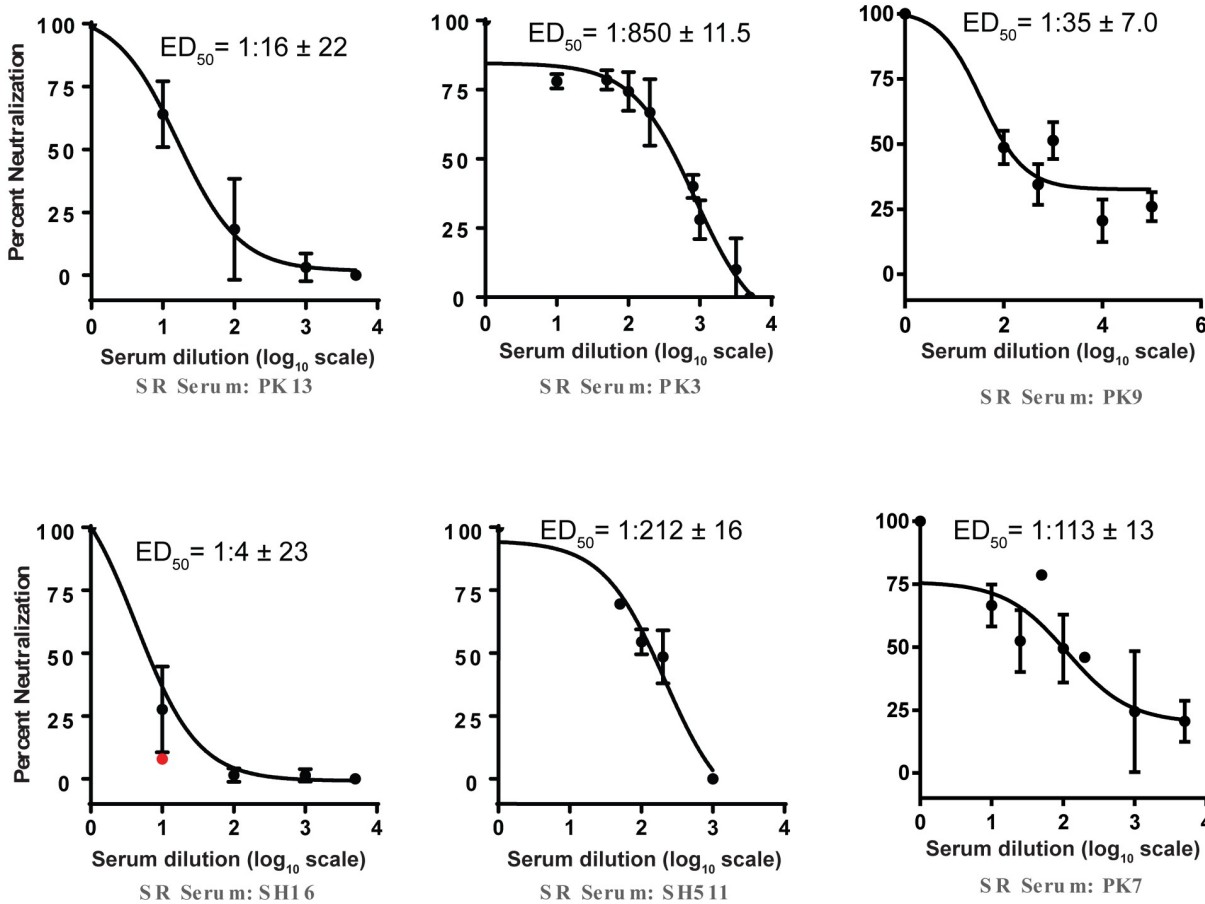

**Fig 4. Dose response curves of selected SR serum samples to measure their ED$_{50}$ values in HCVpp neutralization assay.** Dose response curves showing neutralization of HCVpp by serum samples of spontaneous resolvers. Serum samples at varying dilutions were mixed with HCVpp at 37°C and subsequently added to the wells containing Huh7.5 cells. After 72 h, cells were lysed and luciferase activity was measured. Subsequently, percent neutralization of each dilution was calculated by comparing with the control containing no serum.

For the SR group, we tested HCVpp neutralization of 23 serum samples that were reactive in ELISA along with a few additional non-reactive samples. Twelve of these samples neutralized HCVpp and were subjected to ED$_{50}$ measurement through dose response curves (Fig 4, Table 2). Out of these 12 samples 8 were reactive to the 432-epitope including 7 reactive only to this epitope. The 432-epitope reactive serum samples had higher ED$_{50}$ values as compared to other samples (Table 2) indicating higher titers of 432-epitope reactive antibodies. Three of the HCV neutralizing SR samples were specific only to the 412-epitope; one sample had dual reactivity with the 523-epitope and the 432-epitope, and one (sample SH503) was not reactive to any of the three peptides (Table 2).

## Peptide competition assay

To further clarify if the observed HCV neutralization by the serum samples was due to the linear epitope specific antibodies that were the source of reactivity in ELISA, we performed peptide competition HCV neutralization assay. In these experiments the HCV neutralizing sera reactive to specific linear epitopes were incubated with individual peptides before testing their viral neutralizing activity. Among the 10 CP sera tested in competition assay, 8 samples showed significant decrease in viral neutralization of the sera by the presence of competing

**Table 3. Details of peptide reactivity (epitope specificity) and effect of respective peptides on HCVpp neutralization activity of different CP serum samples.**

| Sample ID | Epitope specificity | HCV neutralizing activity competed by peptides |
|---|---|---|
| 191 | 412, 432, 523 | 412, 432, 523 |
| 193 | 412, 432, 523 | 412, 432, 523 |
| 201 | 412, 432 | 412, 432 |
| 196 | 412, 432, 523 | 432, 523 |
| 226 | 412, 432, 523 | 432, 523 |
| 336 | 412, 432 | 412, 432 |
| 843 | 412 | No effect |
| 2967 | 412, 432 | 412, 432 |
| 574 | 412, 432 | 432 |
| 762 | 412,432 | No effect |

peptides (Table 3). Four of these samples had specificity for the 523-epitope that contains some most conserved residues directly involved in the interactions with CD81 receptor during the viral cellular entry.

For the SR group, 12 serum samples with HCVpp neutralization activity were subjected to competition assay including the one with no ELISA reactivity to any of the peptide. Apart from two serum samples, SH511 and SH19, HCV neutralizing activity of all peptide reactive samples were competed by the presence of respective peptides (Fig 5). Six of these samples had specificity for the 432-epitope and two were specific for the 412-epitope. A sample, SH16 that had shown reactivity to the 432- and the 523-epitopes, was competed by both of the peptides. A neutralizing serum sample without reactivity to any of the peptides was used in competition assay to rule out false positive effects, as its activity was not affected by any of the peptides. Moreover an irrelevant peptide corresponding to a part of the V3 domain of HIV gp120 was also used in competition assays as nonspecific peptide control. Overall, in 73% of the HCV neutralizing serum samples of particular linear epitope reactivity of the SR group the neutralization was significantly affected by the presence of respective peptides. Among these, 75% of the samples had specificity for the 432-epitope. Among all HCV neutralizing SR samples with ELISA reactivity, 54% of the samples showed their dependency primarily on the 432-epitope-targeted antibodies. Moreover, A serum sample showed its HCV neutralization dependency on the antibodies specific for the 523-epitope containing some highly conserved residues, and this epitope has rarely been observed as a linear neutralizing epitope before.

## Discussion

Peptide or small protein subunit based-immunogens as vaccine candidates could be more advantageous over the whole surface protein or attenuated virus owing to specific response of antibodies against smaller immunogens [32]. Viral surface proteins incorporate epitopes of neutralizing, non-neutralizing and interfering antibodies [33,34]. Elicitation of all kinds of antibodies by a vaccine, in particular, interfering antibodies could mask neutralizing epitopes limiting the efficacy of a vaccine [35]. Moreover, surface exposed regions of a viral surface protein exhibit higher sequence variability and antibodies targeting these epitopes are not broadly neutralizing [35]. Sequentially conserved epitopes, which are generally less exposed, can elicit antibodies that are broadly cross reactive [36,37]. In recent years, small peptides corresponding to conserved epitopes of antibodies have been considered as potential vaccine candidates [32,38]. In this study we characterized the conserved linear epitope specificity of NAbs potentially involved in spontaneous viral clearance by detecting the reactivity of the serum samples

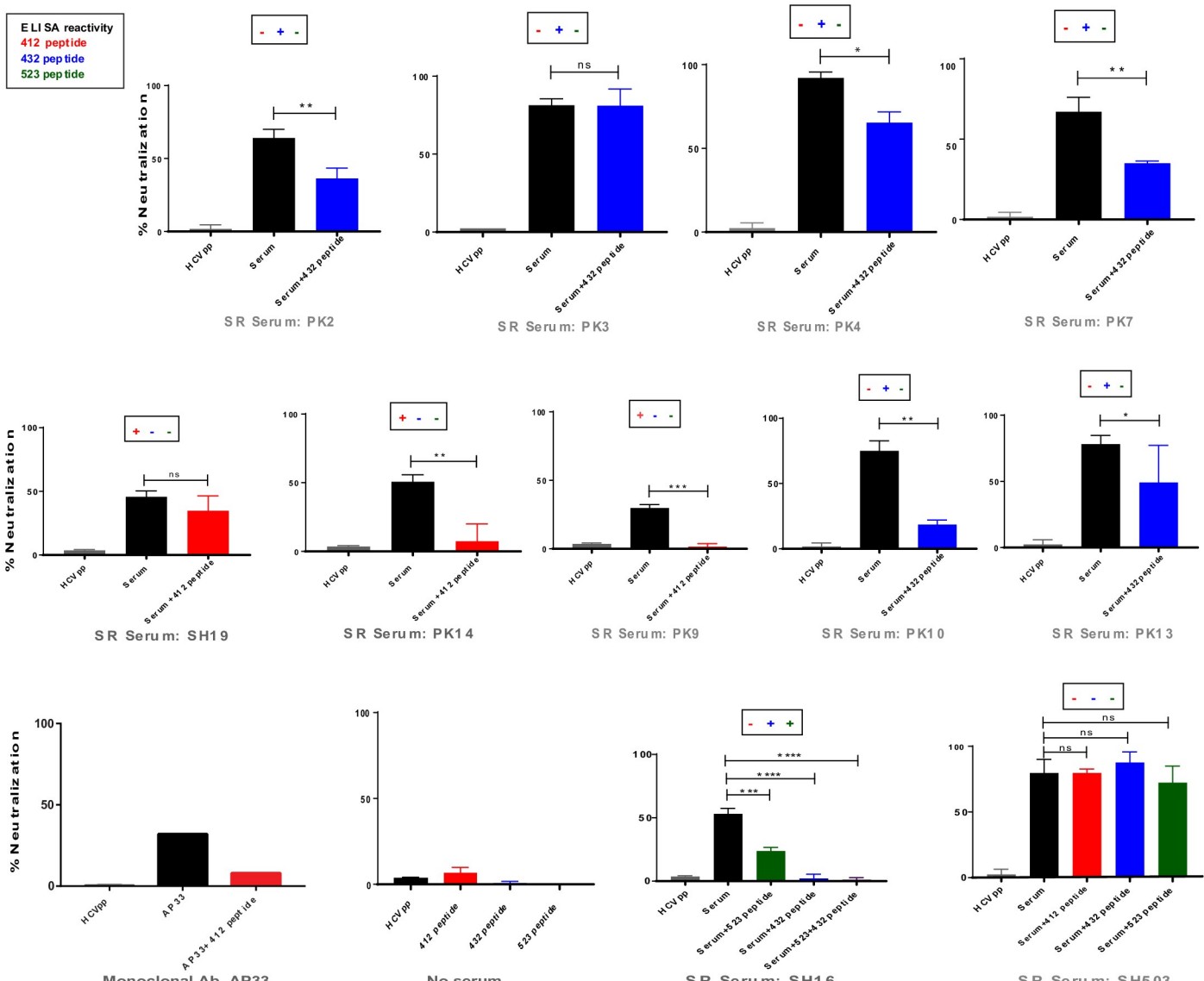

**Fig 5. Effect of peptides corresponding to different conserved linear epitopes on HCVpp neutralization by the SR serum samples.** HCVpp neutralizing serum samples of spontaneous resolvers were incubated at 1:50 dilution with different peptides at 125 μg/ml concentration at 37°C for 2 hr before performing HCVpp neutralization assay. Column bars labeled with HCVpp represent 100% infection; column bars labeled with "Serum" represent HCV neutralization by the respective serum sample in the absence of peptide; and column bars labeled with "Serum+412", "Serum+432" or "Serum+523" represent neutralization by the serum samples after incubation of that serum with respective peptides. Plus and minus signs at the tops of each graph indicates the presence or absence of ELISA reactivity of that serum sample with individual peptides indicated by colors. HCVpp neutralization by the AP33 monoclonal antibody at 1 μg/ml concentration and the effect of the 412 peptide on this neutralization was also determined and shown in the bottom left panel. significance was defined as $^*p \leq 0.05$, $^{**}p \leq 0.01$, $^{***}p \leq 0.001$, $^{****}p \leq 0.0001$.

with peptide epitopes and subsequently evaluating the dependency of serum samples on particular epitope specific antibodies to neutralize HCVpp. In this regard we identified spontaneous resolvers on the basis of positive anti-HCV antibody response and negative viral RNA detection. The diagnostic anti-HCV antibody response in spontaneous resolvers and chronic patients was observed to be comparable indicating these antibodies persist for a long period of time after the clearance of the virus.

For the particular linear epitope specific antibody response, we observed that most of the CP serum samples were reactive in ELISA and to more than one peptide suggesting that antibodies of diverse linear-epitope specificities were present in majority of the chronic patients. On the other hand less than half of the SR samples showed reactivity to any of the three peptides, and most of these were reactive to a single epitope that was primarily the 432-epitope. We do want to mention that direct immobilization of linear peptides on the surface of ELISA plate has potential limitation of false negative outcome due to the conformational restriction of the immobilized peptide. Owing to this limitation, reactivity of a small proportion of antibodies might have missed in our ELISA assay. However, the comparison of the ELISA reactivity of two groups, suggest that only a proportion of spontaneous resolvers produced antibodies targeted against these linear-epitopes and those antibodies were more specific in their reactivity as compared to ones present in chronic patients. Early appearance of neutralizing antibodies directed against a narrow range of envelope epitopes has also been described previously as a key factor in spontaneous HCV clearance [39,40]. In chronic patients, an active infection phase persists for a long period of time therefore high titers of different antibodies of diverse specificities are produced. On the other hand individuals who can clear the virus at acute phase, experience active infection for a short period of time therefore contain low overall antibody titers. However, in our ELISA based analysis, specific linear epitope reactive antibodies were observed in comparable titers in the two groups (Fig 1) suggesting that these linear epitope specific antibodies possibly exist in the larger proportion of the total antibodies in spontaneous-resolvers as compared to in chronic patients, and these antibodies could have role in spontaneous clearance. It has previously been observed that the spontaneous viral clearance by NAbs at the acute phase depends on the early induction of these NAbs at higher titers as such antibodies do not appear at the acute phase of chronic infections [39,40]. Our observation of comparable titers of specific linear epitope targeted antibodies in the two groups of serum samples is in agreement with these findings.

Around half of the ELISA reactive serum samples of each group neutralized HCVpp. This suggested that in roughly half of the cases antibodies reactive to any of the three peptides were non-neutralizing and/or had been interfered by other antibodies to neutralize as several previous studies have reported the masking of neutralizing epitopes on the HCV envelope by interfering non-neutralizing antibodies [41,42]. Overall neutralizing $ED_{50}$ values of the CP serum samples were much higher as compared to those of SR samples. In chronic patients due to the longer exposure several types of antibodies targeting various epitopes are produced. These epitopes include conformational as well as different linear epitopes [34,43–46]. The observed higher $ED_{50}$ values of the CP serum samples can be explained by the presence of a diverse set of antibodies targeting different epitopes on the virus that contribute to HCVpp neutralization.

The neutralizing activity of most of the CP sera was affected by the presence of the competing peptides that had shown reactivity in ELISA. This suggests that antibodies specific for different linear epitopes in the CP sera contributed to the HCV neutralization by these serum samples. Despite the presence of these NAbs targeted against conserved epitopes the infection persisted at chronic phase in these patients. Several previous studies have described that the appearance of broadly cross-reactive antibodies at the early acute phase is critical for the clearance of the virus and appearance of such antibodies at the later stage in chronic progressors cannot overcome the emergence of escape variants [16,39,47]. However, the details of mutations responsible for the escape of neutralization by these conserved epitope specific antibodies remains to be evaluated in the follow up investigation.

In the SR group HCV neutralizing activity of most of the neutralizing serum samples was significantly affected by the presence of competing peptides according to the ELISA

reactivity suggesting HCV neutralization by most of these samples was partially or fully dependent on antibodies specific for respective linear epitope. Among these samples 75% showed their dependency primarily on the 432-epitope-targeted antibodies to neutralize the virus suggesting a potential role of this neutralizing epitope in spontaneous viral clearance. Overall, out of 49 SR serum samples included in this study, 6 (12%) showed their dependency on the 432 linear epitope specific antibodies for HCVpp neutralization while 2 and 1 of the samples respectively were dependent on the 412- and the 523-epitopes targeted antibodies. Several broadly cross-reactive antibodies targeting the 412-epitope have been identified [27]. However, no antibody targeting the 523-region as a linear epitope has been reported earlier but several antibodies specific for conformational epitopes involving residues from the 523-region have been described [48]. The E2 region encompassing aa 432–443 contains highly conserved overlapping linear epitopes and parts of conformational epitopes, and the antibodies targeting this region exhibit the broadest reactivity as compared to the ones specific for other epitopes [49,50]. Keck et al., recently reported the early acute phase appearance of NAbs that target a conformational epitope involving the aa 432–446 region of E2 in an individual who spontaneously resolved three sequential HCV infections [50]. Anne Olbrich et al., recently identified two broadly neutralizing antibodies responsible for spontaneous resolution of the infection at the acute phase, these antibodies specifically recognize a linear epitope (aa483-499) in the central front layer of the E2 glycoprotein. These studies highlight the significance of linear epitopes in general and the region encompassing aa 432–443 of E2 in spontaneous viral clearance [18]. Although, we observed possible role of three conserved linear epitopes targeted NAbs only in a small proportion of individuals, these linear epitopes, in particular the 432-epitope, represent important neutralizing epitopes for spontaneous viral clearance and further studies on structural basis of their recognition by respective antibodies will provide more information for rational vaccine design. There are multiple factors potentially contributing to the spontaneous viral clearance, however in this study we have discussed only the potential contribution of antibodies specific for linear epitopes.

Taken together, we have identified that individuals who spontaneously resolve HCV infection at the acute phase can produce antibodies specific for highly conserved linear epitopes, and those antibodies can play a primary role in the spontaneous clearance of the infection. Among these conserved linear epitopes the E2 region encompassing aa 432–443 was identified as the prominent neutralizing epitope with potential role in spontaneous viral clearance. Further study is needed to delineate the epitope-antibody recognition details using purified monoclonal antibodies as such information would enhance our understanding in rational vaccine design perspective.

## Supporting information

**S1 Fig. Alignment of amino acid sequences of three linear epitopes from different HCV genotypes.** The alignment shows significant amino acid sequence conservation in these regions among different genotypes.
(PDF)

**S2 Fig. VSV neutralization by four different human serum samples.** The data showing no viral neutralization by any of the serum samples at 10-fold dilution. Only the sample 226 showed slight activity.
(PDF)

## Acknowledgments

We are thankful to Dr Charlis M. Rice at the Rockefeller University for providing Huh7.5 cells, to Dr Parizia Farci at NIAID, National Institutes of Health, USA, for providing vectors containing genes encoding HCV envelope and to AIDS Research and Reference Program, Division of AIDS, NIAID, National Institutes of Health, USA for providing backbone plasmid to generate HCVpp. We are also thankful to Punjab AIDS Control Program, Pakistan for providing monthly stipend to Saira Dar.

## Author Contributions

**Conceptualization:** Syed Shahzad-ul-Hussan.

**Data curation:** Asma Ahsan.

**Formal analysis:** Asma Ahsan, Saira Dar, Syed Shahzad-ul-Hussan.

**Funding acquisition:** Syed Shahzad-ul-Hussan.

**Investigation:** Asma Ahsan, Saira Dar, Syed Shahzad-ul-Hussan.

**Methodology:** Fareeha Hassan.

**Project administration:** Syed Shahzad-ul-Hussan.

**Resources:** Fareeha Hassan, Farkhanda Ghafoor, Muhammad Haroon Yousuf.

**Supervision:** Syed Shahzad-ul-Hussan.

**Validation:** Syed Shahzad-ul-Hussan.

**Writing – original draft:** Asma Ahsan.

**Writing – review & editing:** Saira Dar, Syed Shahzad-ul-Hussan.

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
