## [Decision Letter · Decision Letter 0]

25 May 2021

PONE-D-21-08014

Characterization of epitope specificity of antibodies potentially contributing to spontaneous clearance of hepatitis C virus

PLOS ONE

Dear Dr. Shahzad-ul-Hussan,

Thank you for submitting your manuscript to PLOS ONE. After careful consideration, we feel that it has merit but does not fully meet PLOS ONE’s publication criteria as it currently stands. Therefore, we invite you to submit a revised version of the manuscript that addresses the points raised during the review process.

ACADEMIC EDITOR:

I commend the team for such a extensive study and is need of the hour. Please respond with comments (point-by-point) to both the reviewers concerns. Reviewer 1 has raised very important comments and is necessary to fulfill these comments.

We look forward to receiving your revised manuscript.

Kind regards,

Sripathi M Sureban, Ph.D.

Academic Editor

PLOS ONE

Additional Editor Comments:

I commend the team for such a extensive study and is need of the hour. Please respond with comments (point-by-point) to both the reviewers concerns. Reviewer 1 has raised very important comments and is necessary to fulfill these comments.

Journal Requirements:

2. Please provide additional details regarding participant consent. In the ethics statement in the Methods and online submission information, please ensure that you have specified (1) whether consent was informed and (2) what type you obtained (for instance, written or verbal, and if verbal, how it was documented and witnessed).

3. In your Methods section, please provide additional information about the participant recruitment method and the demographic details of your participants. Please ensure you have provided sufficient details to replicate the analyses such as: a) a description of any inclusion/exclusion criteria that were applied to participant recruitment, b) a description of how participants were recruited, and c) descriptions of where participants were recruited and where the research took place.

4. Please ensure you have discussed any potential limitations of your study in the Discussion.

5. To comply with PLOS ONE submission guidelines, in your Methods section, please provide additional information regarding your statistical analyses. For more information on PLOS ONE' expectations for statistical reporting, please see https://journals.plos.org/plosone/s/submission-guidelines.#loc-statistical-reporting.

Reviewers' comments:

Reviewer's Responses to Questions

**Comments to the Author**

1. Is the manuscript technically sound, and do the data support the conclusions?

Reviewer #1: No

Reviewer #2: Partly

2. Has the statistical analysis been performed appropriately and rigorously? 

Reviewer #1: No

Reviewer #2: Yes

3. Have the authors made all data underlying the findings in their manuscript fully available?

Reviewer #1: No

Reviewer #2: No

4. Is the manuscript presented in an intelligible fashion and written in standard English?

Reviewer #1: Yes

Reviewer #2: No

5. Review Comments to the Author

Reviewer #1: Ahsen and colleagues describe an analysis of the epitope specificity of the polyclonal antibody response in patients who spontaneously resolve hepatitis C virus infection, comparing with the antibody response in individuals who progress to chronic infection. The study addresses an important question, as the quality of the antibody response associated with protection is useful for informing vaccine design. However, I have the following concerns with the design and execution of the study:

Major points

1.The design of the study only permits interrogation of the antibody response to linear peptides. As the authors indicate in the Discussion section, it is well established that the majority of neutralizing antibodies isolated from infected humans recognise conformation-dependent epitopes. The approaches used in this study only examine reactivity to three short peptide epitopes, and as such there is the possibility of lack of interrogation of important antibodies binding to conformation-dependent epitopes in these peptide regions on the surface of the E2 protein.

2.The three epitope regions interrogated in the study are all variable to different degrees between different HCV infections. The infecting genotype of HCV influences this sequence variability. While genotype 3 infections are mentioned in the Results, there is no detailed information provided in the Methods about the genotypes of the infecting viruses, either in chronic infections, on in spontaneously resolving infections. Were all of these infections genotype 3 HCV? There also appears to be inconsistencies in the sequences of the selected sequences used for the study. For example, in peptide 412, the sequence RRQLVNTNGSWHINRR is used as the target peptide, but the sequence for genotype 3a indicated in supplementary material is QLINTNGSWHIN. It appears that the peptides were generated as consensus amino acid sequences of all 7 major genotypes of HCV for target peptides. Peptides representing the sequences of genotype 3 strains (if they are the main genotype infecting the study cohort) would be much better for this study. It should also be appreciated that some of the infected individuals will be infected with strains that differ in their amino acid sequence in these critical epitopes, which may influence the reactivity to the selected peptides.

3. Line 88: The synthesis format of the peptides is not described, and it is not clear if the peptides are linear, or branched. Given that the peptides are directly coated onto the assay plates, it is possible that the antibody binding epitopes are not accessible to the antibodies present in the patient samples in this assay format. The authors must comment on the format of the assays used for these analyses. There is the potential for Type II error in the data if the peptides were linear in nature, rather than branched.

4. Line 93: antibody AP33 is indicated to be a positive control for these assays. However, AP33 is a mouse monoclonal antibody, rather than a human antibody, and as such the assay described would not work with the indicated secondary antibody. Did the authors use a humanised version of AP33? If so, this information must be provided, with an appropriate citation.

5. Line 107: The authors should provide information on the strain used for the HCV E1/E2 envelope glycoprotein expression construct. This is extremely important; it is not clear if the neutralization assays were performed with a genotype-matched construct, or if the sequence of the E1/E2 construct matched the amino acid sequence of the peptides used in the ELISA assays.

6. The peptides used in these assays may adopt different conformations depending on the assay format. There is extensive evidence that the peptide representing epitope 412 adopts different conformational configurations, and it is highly likely that in the ELISA format used the peptide conformations will be constrained, while the soluble peptide used for competition assays may be able to adopt different conformations. The authors should comment on this potential issue comparing datasets.

7. Figure 1: the authors should present the ELISA binding data from the uninfected 'healthy' controls in this as an additional panel for each peptide. It is not appropriate for the background cut-off to be set at the mean value of reactivity of six healthy control sera. It would be more appropriate to set the cut-off value at either twice the mean of the negative controls, or mean of controls +2x the standard deviation (if the data fit a Gaussian distribution). With the cut-off values set as presented, many of the ‘positive’ results are likely to be similar to those from the control dataset.

8. Figure 4: the authors describe the use of VSV-G pseudotyped viruses as a control. They should present the data demonstrating that neutralization by serum samples was specific to HCV. Also, healthy, HCV-negative control sera should also be included in the datasets describing neutralization, to demonstrate that the observed neutralization was due to anti-HCV antibodies.

9. Figure 5: The authors should include monoclonal neutralizing antibodies recognising peptides 412 and 434 in these assays to demonstrate that the consensus peptides have the potential to inhibit specific neutralization in each case. I appreciate that neutralizing mAbs recognising peptide 524 may not be available. Even if a mAb for peptide 434 is unavailable, they could perform this experiment for the combination of mAb AP33 and peptide 412.

Minor points

‘Self-resolvers’ should be more accurately referred to as ‘spontaneous resolvers’

Lines 166-171: It is not clear why the figure legend for Figure 1 is presented here.

Lines 187-188: it is not clear why the figure legend for Figure 2 is presented here.

Lines 208-214: it is not clear why the figure legend for Figure 3 is presented here.

Lines 226-232: it is not clear why the figure legend for Figure 4 is presented here.

Figure 4: The third panel on the first row indicates that neutralization is tending to plateau at ~65% neutralization, and at the greatest dilution is trending to a minimum neutralization of ~15%. These data make no biological sense, and are likely to be artefacts of the way the data has been analysed using Graphpad Prism.

Figure 4: the X axis on all panels is labelled as ‘log of serum dilution’, but no logarithmic base is indicated. Presumably they are base 10, but this needs to be indicated.

Line 367: ‘Zhen-Young K et al’ should read ‘Keck et al’

Reviewer #2: This manuscript has the potential to be accepted, but there are a few key problems that need to be explained or rectified before we can move forward and take constructive action. Methods and discussion requires improvement.

6. PLOS authors have the option to publish the peer review history of their article (what does this mean?). If published, this will include your full peer review and any attached files.

Reviewer #1: No

Reviewer #2: No

---

## [Author Response · Author response to Decision Letter 0]

2 Jul 2021

Thanks for the review process, we are submitting the revised version which has been improved according the advice of the reviewers. Early decision would be highly appreciated

---

## [Decision Letter · Decision Letter 1]

23 Jul 2021

PONE-D-21-08014R1

Characterization of linear epitope specificity of antibodies potentially contributing to spontaneous clearance of hepatitis C virus

PLOS ONE

Dear Dr. Shahzad-ul-Hussan,

Thank you for submitting your manuscript to PLOS ONE. After careful consideration, we feel that it has merit but does not fully meet PLOS ONE’s publication criteria as it currently stands. Therefore, we invite you to submit a revised version of the manuscript that addresses the points raised during the review process.

ACADEMIC EDITOR: The manuscript has been extensively improved from its initial version. Reviewer 2 has accepted the manuscript, however, Reviewer 1 has some minor concerns that need to be clarified before considering to accept the manuscript. Please edit and provided point-to-point response to the questions raised.

We look forward to receiving your revised manuscript.

Kind regards,

Sripathi M Sureban, Ph.D.

Academic Editor

PLOS ONE

Journal Requirements:

Additional Editor Comments (if provided):

The manuscript has been extensively improved from its initial version. Reviewer 2 has accepted the manuscript, however, Reviewer 1 has some minor concerns that need to be clarified before considering to accept the manuscript. Please edit and provided point-to-point response to the questions raised.

Reviewers' comments:

Reviewer's Responses to Questions

**Comments to the Author**

1. If the authors have adequately addressed your comments raised in a previous round of review and you feel that this manuscript is now acceptable for publication, you may indicate that here to bypass the “Comments to the Author” section, enter your conflict of interest statement in the “Confidential to Editor” section, and submit your "Accept" recommendation.

Reviewer #1: (No Response)

Reviewer #2: All comments have been addressed

2. Is the manuscript technically sound, and do the data support the conclusions?

Reviewer #1: Partly

Reviewer #2: Yes

3. Has the statistical analysis been performed appropriately and rigorously? 

Reviewer #1: No

Reviewer #2: I Don't Know

4. Have the authors made all data underlying the findings in their manuscript fully available?

Reviewer #1: Yes

Reviewer #2: Yes

5. Is the manuscript presented in an intelligible fashion and written in standard English?

Reviewer #1: Yes

Reviewer #2: Yes

6. Review Comments to the Author

Reviewer #1: In this resubmission the authors have made significant improvements to the quality of the manuscript. This has addressed many of my original concerns, providing additional data to support their conclusions. I have the following comments that need to be addressed:

1. I thank the authors for the rebuttal to my original comment about the format of the ELISA assay and the possibility for false negative results. We and others found that linear peptides derived from HCV E2 directly coated to plastic surfaces were non-reactive for some patient serum samples, but that when the same linear peptides were captured to an ELISA plate using a terminal tag, sero-reactivity could be demonstrated. An alternative solution to this issue is to use branching synthesis when making the peptides to provide additional epitope accessibility (there is no implication that the structure on the surface of E2 is branched) (see Tarr et al, Hepatology 2006 https://doi.org/10.1002/hep.21088; Tarr et al, Journal of General Virology https://doi.org/10.1099/vir.0.83065-0). The authors should mention this potential limitation in their assay format in the Discussion section. The authors cite previous studies that use this method, but it is possible that they also suffer from this limitation.

2. Line 132: the authors have provided the genotype of the strain used for pseudotype generation. However, the neutralization sensitivity of different genotype 1a strains varies widely (Wasilewski, Ray & Bailey, 2016 10.1099/jgv.0.000608). The authors should provide the strain designation of the E1/E2 construct used for these experiments. Use of the H77 strain for these types of experiments is common, and it is widely appreciated that this represents a ‘neutralization-sensitive’ phenotype, that might not represent the neutralization sensitivity of other genotype 1a strains. An indication of the strain would allow the reader to make their own interpretation of the data.

3. Line 293: reference to ‘Table 2’ should be to Table 3.

4. Lines 372-373: It is plausible that the lack of neutralizing response in spontaneous resolvers could be due to the waning antibody response following lack of antigenic stimulation of B cells in these individuals in the period following clearance of the virus. Without a previous RNA-positive test, it is impossible to predict when the spontaneous resolution occurred in these individuals. I appreciate that these patients were identified as being antibody positive in clinical assays, but it would be useful to describe in the Discussion if the signals in these assays were comparable to that achieved in chronic infections, or if the antibody binding signals were low.

5. Figure S1: The authors should revise this figure to represent the most frequent sequences occurring in each genotype (or subgenotype). For instance, for genotype 2b viruses, for sequences deposited on Genbank the sequence for the region 412-423 is rarely ‘QLVNTNGSWHIN’, and is more commonly ‘QLINTNGSWHIN’ of ‘SLINTNGSWHIN’. This impacts on the statement on Line 164 that the selected peptide represented a consensus of all major genotypes of HCV

6. The statistical analysis for Figure 5 used repeated T tests, but there is no indication that a correction for repeated measures was performed. A one-way ANOVA with Bonferroni correction may be more appropriate where multiple comparisons are performed.

Reviewer #2: (No Response)

7. PLOS authors have the option to publish the peer review history of their article (what does this mean?). If published, this will include your full peer review and any attached files.

Reviewer #1: No

Reviewer #2: **Yes: **Dr. R.PARTHIBAN

---

## [Author Response · Author response to Decision Letter 1]

30 Jul 2021

Thanks for the review process to improve the manuscript, we are submitting the revised version which has been revised in light of the reviewers suggestions. I hope it will be accepted now

---

## [Decision Letter · Decision Letter 2]

17 Aug 2021

Characterization of linear epitope specificity of antibodies potentially contributing to spontaneous clearance of hepatitis C virus

PONE-D-21-08014R2

Dear Dr. Shahzad-ul-Hussan,

We’re pleased to inform you that your manuscript has been judged scientifically suitable for publication and will be formally accepted for publication once it meets all outstanding technical requirements.

Kind regards,

Sripathi M Sureban, Ph.D.

Academic Editor

PLOS ONE

Additional Editor Comments (optional):

The manuscript is extensively modified and I congratulate the authors on the good work and I have recommended to accept for publications. Good luck with future projects!

Reviewers' comments:

Reviewer's Responses to Questions

**Comments to the Author**

1. If the authors have adequately addressed your comments raised in a previous round of review and you feel that this manuscript is now acceptable for publication, you may indicate that here to bypass the “Comments to the Author” section, enter your conflict of interest statement in the “Confidential to Editor” section, and submit your "Accept" recommendation.

Reviewer #1: All comments have been addressed

2. Is the manuscript technically sound, and do the data support the conclusions?

Reviewer #1: Yes

3. Has the statistical analysis been performed appropriately and rigorously? 

Reviewer #1: Yes

4. Have the authors made all data underlying the findings in their manuscript fully available?

Reviewer #1: Yes

5. Is the manuscript presented in an intelligible fashion and written in standard English?

Reviewer #1: Yes

6. Review Comments to the Author

Reviewer #1: Thank you for addressing my additional comments on the revised manuscript. I believe the study has been thoroughly improved by the additional data and revisions to the manuscript.

7. PLOS authors have the option to publish the peer review history of their article (what does this mean?). If published, this will include your full peer review and any attached files.

Reviewer #1: No

---

## [Editor Report · Acceptance letter]

19 Aug 2021

PONE-D-21-08014R2 

Characterization of linear epitope specificity of antibodies potentially contributing to spontaneous clearance of hepatitis C virus 

Dear Dr. Shahzad-ul-Hussan:

I'm pleased to inform you that your manuscript has been deemed suitable for publication in PLOS ONE. Congratulations! Your manuscript is now with our production department. 

Kind regards, 

on behalf of

Dr. Sripathi M Sureban 

Academic Editor

PLOS ONE